# First Exit Time Analysis of Stochastic Gradient Descent Under Heavy-Tailed Gradient Noise

**Thanh Huy Nguyen[1], Umut Şimşekli[1,2], Mert Gürbüzbalaban[3], Gaël Richard[1]**
1: LTCI, Télécom Paris, Institut Polytechnique de Paris, France
2: Department of Statistics, University of Oxford, UK
3: Dept. of Management Science and Information Systems, Rutgers Business School, NJ, USA

## Abstract

Stochastic gradient descent (SGD) has been widely used in machine learning due to its computational efficiency and favorable generalization properties. Recently, it has been empirically demonstrated that the gradient noise in several deep learning settings admits a non-Gaussian, heavy-tailed behavior. This suggests that the gradient noise can be modeled by using $\alpha$-stable distributions, a family of heavy-tailed distributions that appear in the generalized central limit theorem. In this context, SGD can be viewed as a discretization of a stochastic differential equation (SDE) driven by a Lévy motion, and the metastability results for this SDE can then be used for illuminating the behavior of SGD, especially in terms of 'preferring wide minima'. While this approach brings a new perspective for analyzing SGD, it is limited in the sense that, due to the time discretization, SGD might admit a significantly different behavior than its continuous-time limit. Intuitively, the behaviors of these two systems are expected to be similar to each other only when the discretization step is sufficiently small; however, to the best of our knowledge, there is no theoretical understanding on how small the step-size should be chosen in order to guarantee that the discretized system inherits the properties of the continuous-time system. In this study, we provide formal theoretical analysis where we derive explicit conditions for the step-size such that the metastability behavior of the discrete-time system is similar to its continuous-time limit. We show that the behaviors of the two systems are indeed similar for small step-sizes and we identify how the error depends on the algorithm and problem parameters. We illustrate our results with simulations on a synthetic model and neural networks.

## 1 Introduction

Stochastic gradient descent (SGD) is one of the most popular algorithms in machine learning due to its scalability to large dimensional problems as well as favorable generalization properties. SGD algorithms are applicable to a broad set of convex and non-convex optimization problems arising in machine learning [1, 2], including deep learning where they have been particularly successful [3, 4, 5]. In deep learning, many key tasks can be formulated as the following non-convex optimization problem:

$$\min_{w \in \mathbb{R}^d} f(w) = \frac{1}{n} \sum_{i=1}^{n} f^{(i)}(w), \tag{1}$$

where $w \in \mathbb{R}^d$ contains the weights for the deep network to estimate, $f^{(i)} : \mathbb{R}^d \mapsto \mathbb{R}$ is the typically non-convex loss function corresponding to the $i$-th data point, and $n$ is the number of data points [6, 7, 5]. SGD iterations consist of

$$W^{k+1} = W^k - \eta \nabla \tilde{f}_k(W^k), \quad k \geq 0, \tag{2}$$

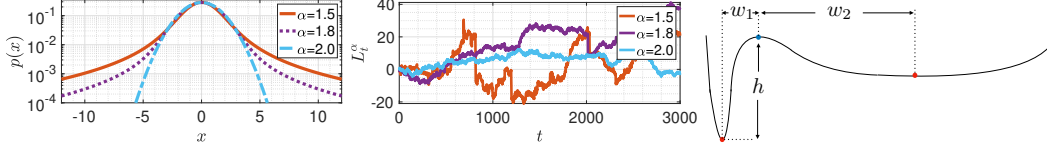

Figure 1: Illustration of $\mathcal{S}\alpha\mathcal{S}$ (left), $L_t^\alpha$ (middle), wide-narrow minima (right).

where $\eta$ is the step-size, $k$ denotes the iterations, $W^0 \in \mathbb{R}^d$ is the initial point, $\nabla \tilde{f}_k(W^k)$ is an unbiased estimator of the actual gradient $\nabla f(W^k)$, estimated from a subset of the component functions $\{f_i\}_{i=1}^n$. In particular, the gradients of the objective are estimated as averages of the form

$$\nabla \tilde{f}_k(W^k) \triangleq \nabla \tilde{f}_{\Omega_k}(W^k) \triangleq \frac{1}{b} \sum_{i \in \Omega_k} \nabla f^{(i)}(W^k), \tag{3}$$

where $\Omega_k \subset \{1, \ldots, n\}$ is a random subset that is drawn with or without replacement at iteration $k$, and $b = |\Omega_k|$ denotes the number of elements in $\Omega_k$ [1].

The popularity and success of SGD in practice have motivated researchers to investigate and analyze the reasons behind; a topic which has been an active research area [6, 4]. One well-known hypothesis [8] that has gained recent popularity (see e.g. [4, 9]) is that among all the local minima lying on the non-convex energy landscape defined by the loss function (1), local minima that lie on wider valleys generalize better compared to sharp valleys, and that SGD is able to converge to the "right local minimum" that generalizes better. This is visualized in Figure 1(right), where the local minimum on the right lies on a wider valley with width $w_2$ compared to the local minimum on the left with width $w_1$ lying in a sharp valley of depth $h$. Interpreting this hypothesis and the structure of the local minima found by SGD clearly requires a deeper understanding of the statistical properties of the gradient noise $Z_k \triangleq \nabla \tilde{f}(W^k) - \nabla f(W^k)$ and its implications on the dynamics of SGD. A number of papers in the literature argue that the noise has Gaussian structure [10, 7, 11, 12, 13, 3]. Under the Gaussian noise assumption, the following continuous-time limit of SGD has been considered in the literature to analyze the behavior of SGD:

$$dW(t) = -\nabla f(W(t))dt + \sqrt{\eta}\sigma dB(t) \tag{4}$$

where $B(t)$ is the standard Brownian motion and $\sigma$ is the noise variance and $\eta$ is the step-size. The Gaussianity of the gradient noise implicitly assumes that the gradient noise has a finite variance with light tails. In a recent study, [6] empirically illustrated that in various deep learning settings, the gradient noise admits a heavy-tail behavior, which suggests that the Gaussian-based approximation is not always appropriate, and furthermore, the heavy-tailed noise could be modeled by a symmetric $\alpha$-stable distribution ($\mathcal{S}\alpha\mathcal{S}(\sigma)$). Here, $\alpha \in (0, 2]$ is called the *tail-index* and characterizes the heavy-tailedness of the distribution and $\sigma$ is a scale parameter that will be formally defined in Section 2. This $\alpha$-stable model generalizes the Gaussian model in the sense that $\alpha = 2$ reduces to the Gaussian model, whereas smaller values of $\alpha$ quantify the heavy-tailedness of the gradient noise (see Figure 1(left)). Under this noise model, the resulting continuous-time limit of SGD becomes [6]:

$$dW(t) = -\nabla f(W(t))dt + \eta^{\frac{\alpha-1}{\alpha}}\sigma dL^\alpha(t), \tag{5}$$

where $L^\alpha(t)$ is the $d$-dimensional $\alpha$-stable Lévy motion with independent components (which will be formally defined in Section 2). This process has also been investigated for Bayesian posterior sampling [14] and global non-convex optimization [15].

The sample paths of the Lévy-driven SDE (5) have a fundamentally different behavior than the ones of Brownian motion driven dynamics (4). This difference is mainly originated by the fact that, unlike the Brownian motion which has almost surely continuous sample paths, the Lévy motion can have discontinuities, which are also called 'jumps' [16] (cf. Figure 1(middle)). This fundamental difference becomes more prominent in the metastability properties of the SDE (5): consider a basin in which a particle is initialized and undergoes fluctuations continually. The particle persists in the basin for a long time before exiting it by the influence of fluctuations. This relative instability phenomenon is described by the term 'metastability'.

More formally, the metastability studies consider the case where $W(0)$ is initialized in a basin and analyze the minimum time $t$ such that $W(t)$ *exits* that basin. It has been shown that when $\alpha < 2$ (i.e. the noise has a heavy-tailed component), this so called *first exit time* only depends on the *width* of the

basin and the value of $\alpha$, and it does not depend on the height of the basin [17, 18, 19]. The empirical results in [6] showed that, in various deep learning settings the estimated tail index $\alpha$ is significantly smaller than 2, suggesting that the metastability results can be used as a proxy for understanding the dynamics of SGD in discrete time, especially to shed more light on the hypothesis that SGD prefers wide minima.

While this approach brings a new perspective for analyzing SGD, approximating SGD as a continuous-time approach might not be accurate for any step-size $\eta$, and some theoretical concerns have already been raised for the validity of such approximations [20]. Intuitively, one can expect that the metastable behavior of SGD would be similar to the behavior of its continuous-time limit only when the discretization step-size is small enough. Even though some theoretical results have been recently established for the discretizations of SDEs driven by Brownian motion [21], it is not clear that how the discretized Lévy SDEs behave in terms of metastability.

In this study, we provide formal theoretical analyses where we derive explicit conditions for the step-size such that the metastability behavior of the discrete-time system (7) is guaranteed to be close to its continuous-time limit (6). More precisely, we consider a stochastic differential equation with both a Brownian term and a Lévy term, and its Euler discretization as follows [22]:

$$\mathrm{d}W(t) = -\nabla f(W(t-))\mathrm{d}t + \varepsilon\sigma\mathrm{d}B(t) + \varepsilon\mathrm{d}L^\alpha(t) \qquad (6)$$

$$W^{k+1} = W^k - \eta\nabla f(W^k) + \varepsilon\sigma\eta^{1/2}\xi_k + \varepsilon\eta^{1/\alpha}\zeta_k, \qquad (7)$$

with independent and identically distributed (i.i.d.) variables $\xi_k \sim \mathcal{N}(0, I)$ where $I$ is the identity matrix, the components of $\zeta_k$ are i.i.d with $\mathcal{S}\alpha\mathcal{S}(1)$ distribution, and $\varepsilon$ is the amplitude of the noise. This dynamics includes (4) and (5) as special cases. Here, we choose $\sigma$ as a scalar for convenience; however, our analyses can be extended to the case where $\sigma$ is a function of $W(t)$.

Understanding the metastability behavior of SGD modeled by these dynamics requires understanding how long it takes for the continuous-time process $W(t)$ given by (6) and its discretization $W^k$ (7) to exit a neighborhood of a local minimum $\bar{w}$, if it is started in that neighborhood. For this purpose, for any given local minimum $\bar{w}$ of $f$ and $a > 0$, we define the following set

$$A \triangleq \left\{ (w^1, \ldots, w^K) \in \mathbb{R}^d \times \ldots \times \mathbb{R}^d : \max_{k \leq K} \|w^k - \bar{w}\| \leq a \right\}, \qquad (8)$$

which is the set of $K$ points in $\mathbb{R}^d$, each at a distance of at most $a$ from the local minimum $\bar{w}$. We formally define the *first exit times*, respectively for $W(t)$ and $W^k$ as follows:

$$\tau_{\xi,a}(\varepsilon) \triangleq \inf\{t \geq 0 : \|W(t) - \bar{w}\| \notin [0, a + \xi]\}, \qquad (9)$$

$$\bar{\tau}_{\xi,a}(\varepsilon) \triangleq \inf\{k \in \mathbb{N} : \|W^k - \bar{w}\| \notin [0, a + \xi]\}, \qquad (10)$$

where the processes are initialized at $W(0) \equiv W^0$ such that $\|W(0) - \bar{w}\| \in [0, a]$. Our main result (Theorem 2) shows that with sufficiently small discretization step $\eta$, the probability of exiting a given neighborhood of the local optimum at a fixed time $t$ of the discretization process approximates that of the continuous process. This result also provides an explicit condition for the step-size, which explains certain impacts of the other parameters of the problem, such as dimension $d$, noise amplitude $\varepsilon$, variance of Gaussian noise $\sigma$, towards the similarity of the discretization and continuous processes. We validate our theory on a synthetic model and neural networks.

**Notations.** For $z > 0$, the gamma function is defined as $\Gamma(z) \triangleq \int_0^\infty x^{z-1}e^{-x}\mathrm{d}x$. For any Borel probability measures $\mu$ and $\nu$ with domain $\Omega$, the total variation (TV) distance is defined as follows: $\|\mu - \nu\|_{TV} \triangleq 2\sup_{A \in \mathcal{B}(\Omega)} |\mu(A) - \nu(A)|$, where $\mathcal{B}(\Omega)$ denotes the Borel subsets of $\Omega$.

## 2   Technical Background

**Symmetric $\alpha$-stable distributions.** The $\mathcal{S}\alpha\mathcal{S}$ distribution is a generalization of a centered Gaussian distribution where $\alpha \in (0, 2]$ is called the *tail index*, a parameter that determines the amount of heavy-tailedness. We say that $X \sim \mathcal{S}\alpha\mathcal{S}(\sigma)$, if its characteristic function $\mathbb{E}[e^{i\omega X}] = e^{-|\sigma|\omega^\alpha}$ where $\sigma \in (0, \infty)$ is called the *scale parameter*. In the special case, when $\alpha = 2$, $\mathcal{S}\alpha\mathcal{S}(\sigma)$ reduces to the normal distribution $\mathcal{N}(0, 2\sigma^2)$. A crucial property of the $\alpha$-stable distributions is that, when $X \sim \mathcal{S}\alpha\mathcal{S}(\sigma)$ with $\alpha < 2$, the moment $\mathbb{E}[|X|^p]$ is finite if and only if $p < \alpha$, which implies that

$\mathcal{S}\alpha\mathcal{S}$ has infinite variance as soon as $\alpha < 2$. While the probability density function does not have closed form analytical expression except for a few special cases of $\mathcal{S}\alpha\mathcal{S}$ (e.g. $\alpha = 2$: Gaussian, $\alpha = 1$: Cauchy), it is computationally easy to draw random samples from it by using the method proposed in [23].

**Lévy processes and SDEs driven by Lévy motions.** The standard $\alpha$-stable Lévy motion on the real line is the unique process satisfying the following properties [22]:

  $(i)$ For any $0 \le t_1 < t_2 < t_2 < \cdots < t_N$, its increments $L^\alpha_{t_{i+1}} - L^\alpha_{t_i}$ are independent for $i = 1, 2, \ldots, N$ and $L^\alpha_0 = 0$ almost surely.

  $(ii)$ $L^\alpha_{t-s}$ and $L^\alpha_t - L^\alpha_s$ have the same distribution $\mathcal{S}\alpha\mathcal{S}\left((t-s)^{1/\alpha}\right)$ for any $t > s$.

  $(iii)$ $L^\alpha_t$ is continuous in probability: $\forall \delta > 0$ and $s \ge 0$, $\mathbb{P}(|L^\alpha_t - L^\alpha_s| > \delta) \to 0$ as $t \to s$.

When $\alpha = 2$, $L^\alpha_t$ reduces to a scaled version of the standard Brownian motion $\sqrt{2}B_t$. Since $L^\alpha_t$ for $\alpha < 2$ is only continuous in probability, it can incur a countable number of discontinuities at random times, which makes is fundamentally different from the Brownian motion that has almost surely continuous paths.

The $d$-dimensional Lévy motion with independent components is a stochastic process on $\mathbb{R}^d$ where each coordinate corresponds to an independent scalar Lévy motion. Stochastic processes based on Lévy motion such as (5) and their mathematical properties have also been studied in the literature, we refer the reader to [24, 16] for details.

**First Exit Times of Continuous-Time Lévy Stable SDEs.** Due to the discontinuities of the Lévy-driven SDEs, their metastability behaviors also differ significantly from their Brownian counterparts. In this section, we will briefly mention important theoretical results about the SDE given in (6).

For simplicity, let us consider the SDE (6) in dimension one, i.e. $d = 1$. In a relatively recent study [17], the authors considered this SDE, where the potential function $f$ is required to have a non-degenerate global minimum at the origin, and they proved the following theorem.

**Theorem 1** ([17]). *Consider the SDE (6) in dimension $d = 1$ and assume that it has a unique strong solution. Assume further that the objective $f$ has a global minimum at zero, satisfying the conditions $f'(x)x \ge 0$, $f(0) = 0$, $f'(x) = 0$ if and only if $x = 0$, and $f''(0) = M > 0$. Then, there exist positive constants $\varepsilon_0$, $\gamma$, $\delta$, and $C > 0$ such that for $0 < \varepsilon \le \varepsilon_0$, the following holds:*

$$\mathrm{e}^{-u\varepsilon^\alpha \frac{\theta}{\alpha}(1+C\varepsilon^\delta)}(1 - C\varepsilon^\delta) \le \mathbb{P}(\tau_{0,a}(\varepsilon) > u) \le \mathrm{e}^{-u\varepsilon^\alpha \frac{\theta}{\alpha}(1-C\varepsilon^\delta)}(1 + C\varepsilon^\delta) \quad (11)$$

*for all $W(0)$ initialized uniformly in $[-a + \varepsilon^\gamma, a - \varepsilon^\gamma]$ and $u \ge 0$, where $\theta = \frac{2}{a^\alpha}$. Consequently,*

$$\mathbb{E}[\tau_{0,a}(\varepsilon)] = \frac{\alpha}{2}\frac{a^\alpha}{\varepsilon^\alpha}(1 + \mathcal{O}(\varepsilon^\delta)), \quad \text{for all W(0) initialized uniformly in } [-a + \varepsilon^\gamma, a - \varepsilon^\gamma]. \quad (12)$$

This result indicates that the first exit time of $W(t)$ needs only polynomial time with respect to the *width* of the basin and it does not depend on the depth of the basin, whereas Brownian systems need exponential time in the height of the basin in order to exit from the basin [25, 18]. This difference is mainly due to the discontinuities of the Lévy motion, which enables it to 'jump out' of the basin, whereas the Brownian SDEs need to 'climb' the basin due to their continuity. Consequently, given that the gradient noise exhibits similar heavy-tailed behavior to an $\alpha$-stable distributed random variable, this result can be considered as a proxy to understand the wide-minima behavior of SGD.

We note that this result has already been extended to $\mathbb{R}^d$ in [19]. Extension to state dependent noise has also been obtained in [26]. We also note that the metastability phenomenon is closely related to the spectral gap of the forward operator corresponding to the SDE dynamics (see e.g. [25]) and it is known that this quantity scales like $\mathcal{O}(\varepsilon^\alpha)$ for $\varepsilon$ small which determines the dependency to $\varepsilon$ in the first term of the exit time (12) due to Kramer's Law [27, 28]. Burghoff and Pavlyukevich [28] showed that similar scaling in $\varepsilon$ for the spectral gap would hold if we were to restrict the SDE dynamics to a discrete grid with a small enough grid size.

## 3 Assumptions and the Main Result

In this study, our main goal is to obtain an explicit condition on the step-size, such that the first exit time of the continuous-time process $\tau_{\xi,a}(\varepsilon)$ (9) would be similar to the first exit time of its Euler discretization $\bar{\tau}_{\xi,a}(\varepsilon)$ (10).

We first state our assumptions.

**A 1.** *The SDE* (6) *admits a unique strong solution.*

**A 2.** *The process $\phi_t \triangleq -\frac{b(W)+\nabla f(W(t))}{\varepsilon\sigma}$ satisfies $\mathbb{E}\exp\left(\frac{1}{2}\int_0^T \phi_t^2 \mathrm{d}t\right) < \infty$.*

**A 3.** *The gradient of $f$ is $\gamma$-Hölder continuous with $\frac{1}{2} < \gamma < \min\{\frac{1}{\sqrt{2}}, \frac{\alpha}{2}\}$:*

$$\|\nabla f(x) - \nabla f(y)\| \le M\|x-y\|^\gamma, \qquad \forall x, y \in \mathbb{R}^d.$$

**A 4.** *The gradient of $f$ satisfies the following assumption: $\|\nabla f(0)\| \le B$.*

**A 5.** *For some $m > 0$ and $b \ge 0$, $f$ is $(m, b, \gamma)$-dissipative: $\langle x, \nabla f(x)\rangle \ge m\|x\|^{1+\gamma} - b$, $\forall x \in \mathbb{R}^d$.*

We note that, as opposed to the theory of SDEs driven by Brownian motion, the theory of Lévy-driven SDEs is still an active research field where even the existence of solutions with general drift functions is not well-established and the main contributions have appeared in the last decade [29, 30]. Therefore, **A**1 has been a common assumption in stochastic analysis, e.g. [17, 19, 31]. Nevertheless, existence and uniqueness results have been very recently established in [30] for SDEs with bounded Hölder drifts. Therefore **A**1 and **A**2 directly hold for bounded gradients and extending this result to Hölder and dissipative drifts is out of the scope of this study. On the other hand, the assumptions **A**3-**A**5 are standard conditions, which are often considered in non-convex optimization algorithms that are based on discretization of diffusions [32, 33, 34, 35, 36, 37, 38].

Now, we identify an explicit condition for the step-size, which is one of our main contributions.

**A 6.** *For a given $\delta > 0$, $t = K\eta$, and for some $C > 0$, the step-size satisfies the following condition:*

$$0 < \eta \le \min\left\{1, \frac{m}{M^2}, \left(\frac{\delta^2}{2K_1 t^2}\right)^{\frac{1}{\gamma^2+2\gamma-1}}, \left(\frac{\delta^2}{2K_2 t^2}\right)^{\frac{1}{2\gamma}}, \left(\frac{\delta^2}{2K_3 t^2}\right)^{\frac{\alpha}{2\gamma}}, \left(\frac{\delta^2}{2K_4 t^2}\right)^{\frac{1}{\gamma}}\right\},$$

*where $\varepsilon$ is as in* (7)*, the constants $m, M, b$ are defined by* **A**3*–* **A**5 *and*

$$K_1 = \mathcal{O}(d\varepsilon^{2\gamma^2-2}), \quad K_2 = \mathcal{O}(\varepsilon^{-2}), \quad K_3 = \mathcal{O}(d^{2\gamma}\varepsilon^{2\gamma-2}), \quad K_4 = \mathcal{O}(d^{2\gamma}\varepsilon^{2\gamma-2}).$$

**A** 6 will be stated in more details in the supplementary document. We now present our main result, its proof can be found in the supplementary material.

**Theorem 2.** *Under assumptions* **A**1*-* **A**6*, the following inequality holds:*

$$\mathbb{P}[\tau_{-\xi,a}(\varepsilon) > K\eta] - C_{K,\eta,\varepsilon,d,\xi} - \delta \le \mathbb{P}[\bar\tau_{0,a}(\varepsilon) > K] \le \mathbb{P}[\tau_{\xi,a}(\varepsilon) > K\eta] + C_{K,\eta,\varepsilon,d,\xi} + \delta,$$

*where,*

$$C_{K,\eta,\varepsilon,d,\xi} \triangleq \frac{C_1(K\eta(d\varepsilon+1)+1)^\gamma e^{M\eta}M\eta}{\xi} + 1 - \left(1 - Cde^{-\xi^2 e^{-2M\eta}(\varepsilon\sigma)^{-2}/(16d\eta)}\right)^K$$

$$+ 1 - \left(1 - C_\alpha d^{1+\alpha/2}\eta e^{\alpha M\eta}\varepsilon^\alpha \xi^{-\alpha}\right)^K,$$

*for some constants $C_1, C_\alpha$ and $C$ that does not depend on $\eta$ or $\varepsilon$, $M$ is given by* **A**3 *and $\varepsilon$ is as in* (6)–(7)*.*

**Remark.** Theorem 2 enables the use of the metastability results for Lévy-driven SDEs for their discretized counterpart, which is our most important contribution.

**Exit time versus problem parameters.** In Theorem 2, if we let $\eta$ go to zero for any $\delta$ fixed, the constant $C_{K,\eta,\varepsilon,d,\xi}$ will also go to zero, and since $\delta$ can be chosen arbitrarily small, this implies that the probability of the first exit time for the discrete process and the continuous process will approach each other when the step-size gets smaller, as expected. If instead, we decrease $d$ or $\varepsilon$, the quantity $C_{K,\eta,\varepsilon,d,\xi}$ also decreases monotonically, but it does not go to zero due to the first term in the expression of $C_{K,\eta,\varepsilon,d,\xi}$.

**Exit time versus width of local minima.** Popular activation functions used in deep learning such as ReLU functions are almost everywhere differentiable and therefore the cost function has a well-defined Hessian almost everywhere (see e.g. [39]). The eigenvalues of the Hessian of the objective near local minima have also been studied in the literature (see e.g. [40, 41]). If the Hessian around a local minimum is positive definite, the conditions for the multi-dimensional version of Theorem 1 in [19]) are satisfied locally around a local minimum. For local minima lying in wider valleys, the parameter $a$ can be taken to be larger; in which case the expected exit time $\mathbb{E}\tau_{0,a}(\varepsilon) \sim \mathcal{O}(a^\alpha)$ will be larger by the formula (12). In other words, the SDE (5) spends more time to exit wider valleys. Theorem 2 shows that SGD modeled by the discretization of this SDE will also inherit a similar behavior if the step-size satisfies the conditions we provide.

# 4 Proof Overview

Relating the first exit times for $W(t)$ and $W^k$ often requires obtaining bounds on the distance between $W(k\eta)$ and $W^k$. In particular if $\|W^k - W(k\eta)\|$ is small with high probability, then we expect that their first exit times from the set $A$ will be close to each other as well with high probability.

For objective functions with bounded gradients, in order to relate $\tau_{\xi,a}(\varepsilon)$ to $\bar{\tau}_{\xi,a}(\varepsilon)$, one can attempt to use the strong convergence of the Euler scheme (cf. [42] Proposition 1): $\lim_{\eta \to 0} \mathbb{E}\|W^k - W(k\eta)\| = 0$. By using Markov's inequality, this result implies convergence in probability: for any $\delta > 0$ and $\varepsilon > 0$, there exists $\eta$ such that $\mathbb{P}(\|W^k - W(k\eta)\| > \varepsilon) < \delta/2$. Then, if $W(k\eta) \in A$ then one of the following events must happen:

1. $W^k \in A$,
2. $W^k \notin A$ and $\|W^k - W(k\eta)\| > \epsilon$ (with probability less than $\delta/2$),
3. $W^k \notin A$ and distance from $W^k$ to $A$ is at most $\varepsilon$ (with probability less than $\delta/2$).

By using this observation, we obtain: $\mathbb{P}[W(k\eta) \in A] \leq \mathbb{P}[W^k \in A] + \delta$. Even though we could use this result in order to relate $\tau_{\xi,a}(\varepsilon)$ to $\bar{\tau}_{\xi,a}(\varepsilon)$, this approach would not yield a meaningful condition for $\eta$ since the bounds for the strong error $\mathbb{E}\|W^k - W(k\eta)\|$ often grows exponentially in general with $k$, which means $\eta$ should be chosen exponentially small for a given $k$. Therefore, in our strategy, we choose a different path where we do not use the strong convergence of the Euler scheme.

Our proof strategy is inspired by the recent study [21], where the authors analyze the empirical metastability of the Langevin equation which is driven by a Brownian motion. However, unlike the Brownian case that [21] was based on, some of the tools for analyzing Brownian SDEs do not exist for Lévy-driven SDEs, which increases the difficulty of our task.

We first define a *linearly interpolated* version of the discrete-time process $\{W^k\}_{k \in \mathbb{N}_+}$, which will be useful in our analysis, given as follows:

$$d\hat{W}(t) = b(\hat{W})dt + \varepsilon\sigma dB(t) + \varepsilon dL^\alpha(t), \tag{13}$$

where $\hat{W} \equiv \{\hat{W}(t)\}_{t \geq 0}$ denotes the whole process and the *drift* function $b(\hat{W})$ is chosen as follows:

$$b(\hat{W}) \triangleq -\sum_{k=0}^{\infty} \nabla f(\hat{W}(k\eta)) \mathbb{I}_{[k\eta, (k+1)\eta)}(t).$$

Here, $\mathbb{I}$ denotes the indicator function, i.e. $\mathbb{I}_S(x) = 1$ if $x \in S$ and $\mathbb{I}_S(x) = 0$ if $x \notin S$. It is easy to verify that $\hat{W}(k\eta) = W^k$ for all $k \in \mathbb{N}_+$ [43, 32].

In our approach, we start by developing a Girsanov-like change of measures [24] to express the Kullback-Leibler (KL) divergence between $\mu_t$ and $\hat{\mu}_t$, which is defined as follows:

$$\mathrm{KL}(\hat{\mu}_t, \mu_t) \triangleq \int \log \frac{d\hat{\mu}_t}{d\mu_t} d\hat{\mu}_t,$$

where $\mu_t$ denotes the law of $\{W(s)\}_{s \in [0,t]}$, $\hat{\mu}_t$ denotes the law of $\{\hat{W}(s)\}_{s \in [0,t]}$, and $d\mu_t/d\hat{\mu}_t$ is the Radon–Nikodym derivative of $\mu_t$ with respect to $\hat{\mu}_t$. Here, we require **A2** for the existence of a Girsanov transform between $\hat{\mu}_t$ and $\mu_t$ and for establishing an explicit formula for the transform. In the supplementary document, we show that the KL divergence between $\mu_t$ and $\hat{\mu}_t$ can be written as:

$$\mathrm{KL}(\hat{\mu}_t, \mu_t) = \frac{1}{2\varepsilon^2\sigma^2} \mathbb{E}\left[\int_0^t \|b(\hat{W}) + \nabla f(\hat{W}(s))\|^2 ds\right]. \tag{14}$$

While this result has been known for SDEs driven by Brownian motion [16], none of the references we are aware of expressed the KL divergence as in (14). We also note that one of the key reasons that allows us to obtain (14) is the presence of the Brownian motion in (6), i.e. $\sigma > 0$. For $\sigma = 0$ such a measure transformation cannot be performed [44].

In the next result, we show that if the step-size is chosen sufficiently small, the KL divergence between $\mu_t$ and $\hat{\mu}_t$ is bounded.

**Theorem 3.** *Assume that the conditions **A1-A6** hold. Then the following inequality holds:*

$$\mathrm{KL}(\hat{\mu}_t, \mu_t) \leq 2\delta^2.$$

The proof technique is similar to the approach of [43, 32, 15]: The idea is to divide the integral in (14) into smaller pieces and bounding each piece separately. Once we obtain a bound on KL, by using an optimal coupling argument, the data processing inequality, and Pinsker's inequality, we obtain a bound for the total variation (TV) distance between $\mu_t$ and $\hat{\mu}_t$ as follows:

$$\mathbb{P}_{\mathbf{M}}[(W(\eta),\ldots,W(K\eta)) \neq (\hat{W}(\eta),\ldots,\hat{W}(K\eta))] \leq \|\mu_{K\eta} - \hat{\mu}_{K\eta}\|_{TV} \leq \left(\frac{1}{2}\mathrm{KL}(\hat{\mu}_{K\eta},\mu_{K\eta})\right)^{\frac{1}{2}}.$$

where the TV distance is defined in Section 1. Besides, $\mathbf{M}$ denotes the optimal coupling between $\{W(s)\}_{s\in[0,K\eta]}$ and $\{\hat{W}(s)\}_{s\in[0,K\eta]}$, i.e., the joint probability measure of $\{W(s)\}_{s\in[0,K\eta]}$ and $\{\hat{W}(s)\}_{s\in[0,K\eta]}$, which satisfies the following identity [45]:

$$\mathbb{P}_{\mathbf{M}}[\{W(s)\}_{s\in[0,K\eta]} \neq \{\hat{W}(s)\}_{s\in[0,K\eta]}] = \|\mu_{K\eta} - \hat{\mu}_{K\eta}\|_{TV}.$$

Combined with Theorem 3, this inequality implies the following useful result:

$$\mathbb{P}[(W(\eta),\ldots,W(K\eta)) \in A] - \delta \leq \mathbb{P}[\bar{\tau}_{0,a}(\varepsilon) > K] \leq \mathbb{P}[(W(\eta),\ldots,W(K\eta)) \in A] + \delta \quad (15)$$

where we used the fact that the event $(\hat{W}(\eta),\ldots,\hat{W}(K\eta)) \in A$ is equivalent to the event $(\bar{\tau}_{0,a}(\varepsilon) > K)$. The remaining task is to relate the probability $\mathbb{P}[(W(\eta),\ldots,W(K\eta)) \in A]$ to $\mathbb{P}[\tau_{\xi,a}(\varepsilon) > K\eta]$. The event $(W(\eta),\ldots,W(K\eta)) \in A$ ensures that the process $W(t)$ does not leave the set $A$ when $t = \eta,\ldots,K\eta$; however, it does not indicate that the process remains in $A$ when $t \in (k\eta,(k+1)\eta)$. In order to have a control over the whole process, we introduce the following event:

$$B \triangleq \left\{ \max_{0 \leq k \leq K-1} \sup_{t\in[k\eta,(k+1)\eta]} \|W(t) - W(k\eta)\| \leq \xi \right\},$$

such that the event $[(W(\eta),\ldots,W(K\eta)) \in A] \cap B$ ensures that the process stays close to $A$ for the whole time. By using this event, we can obtain the following inequalities:

$$\mathbb{P}[(W(\eta),\ldots,W(K\eta)) \in A] \leq \mathbb{P}[(W(\eta),\ldots,W(K\eta)) \in A \cap B] + \mathbb{P}[(W(\eta),\ldots,W(K\eta)) \in B^c]$$
$$= \mathbb{P}[\tau_{\xi,a}(\varepsilon) > K\eta] + \mathbb{P}[(W(\eta),\ldots,W(K\eta)) \in B^c].$$

By using the same approach, we can obtain a lower bound on $\mathbb{P}[(W(\eta),\ldots,W(K\eta)) \in A]$ as well. Hence, our final task reduces to bounding the term $\mathbb{P}[(W(\eta),\ldots,W(K\eta)) \in B^c]$, which we perform by using the weak reflection principles of Lévy processes [46]. This finally yields Theorem 2.

## 5  Numerical Illustration

**Synthetic data.**  To illustrate our results, we first conduct experiments on a synthetic problem, where the cost function is set to $f(x) = \frac{1}{2}\|x\|^2$. This corresponds to an Ornstein-Uhlenbeck-type process, which is commonly considered in metastability analyses [22]. This process locally satisfies the conditions **A**1-**A**5.

Since we cannot directly simulate the continuous-time process, we consider the stochastic process sampled from (7) with sufficiently small step-size as an approximation of the continuous scheme. Thus, we organize the experiments as follows. We first choose a very small step-size, i.e. $\eta = 10^{-10}$. Starting from an initial point $W^0$ satisfying $\|W^0\| < a$, we iterate (7) until we find the first $K$ such that $\|W^K\| > a$. We repeat this experiment 100 times, then we take the average $K\eta$ as the 'ground-truth' first exit time. We continue the experiments by calculating the first exit times for larger step-sizes (each repeated 100 times), and compute their distances to the ground truth.

The results for this experiment are shown in Figure 2. By Theorem 2, the distance between the first exit times of the discretization and the continuous processes depends on two terms $C_{K,\eta,\varepsilon,d,\bar{\varepsilon}}$ and $\delta$, which are used for explaining our experimental results.

We observe from Figure 2(a) that the error to the ground-truth first exit time is an increasing function of $\eta$, which directly matches our theoretical result. Figure 2(b) shows that, with small noise limit (e.g., in our settings, $\varepsilon < 1$ versus $\eta \approx 10^{-8}$), the error decreases with the parameter $\varepsilon$. By **A**6, with increased $\varepsilon$, we have the term $\delta$ to be reduced. On the other hand, $C_{K,\eta,\varepsilon,d,\bar{\varepsilon}}$ increases with $\varepsilon$. However, at small noise limit, this effect is dominated by the decrease of $\delta$, that makes the error decrease overall. The decreasing speed then decelerates with larger $\varepsilon$, since, the product $\varepsilon\eta$ becomes so large that the increase of $C_{K,\eta,\varepsilon,d,\bar{\varepsilon}}$ starts to dominate the decrease of $\delta$. Thus, it suggests that for

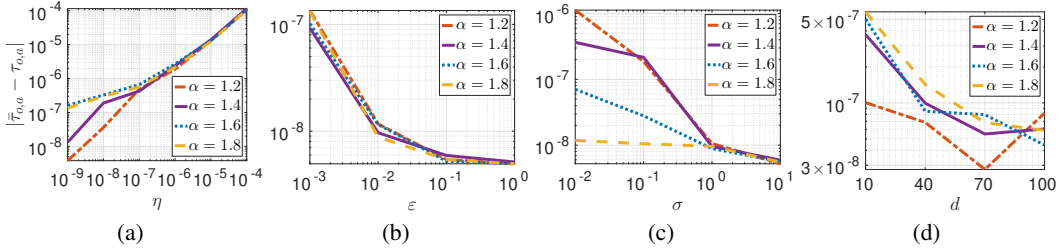

Figure 2: Results of the synthetic experiments.

a large $\varepsilon$, a very small step-size $\eta$ would be required for reducing the distance between the first exit times of the processes. In Figure 2(c), the error decreases when the variance $\sigma$ increases. The reason for the performance is the same as in (b), and can be explained by considering the expression of $\delta$ and $C_{K,\eta,\varepsilon,d,\bar{\varepsilon}}$ in the conclusion of Theorem 2.

In Figure 2(d), for small dimension, with the same exit time interval, when we increase $d$, both processes escape the interval earlier, with smaller exit times. Hence, the distance between their exit times becomes smaller. With larger $d$, the increasing effect of $\delta$ and $C_{K,\eta,\varepsilon,d,\bar{\varepsilon}}$ starts to dominate the above 'early-escape' effect, thus, the decreasing speed of the error diminish. We observe that the error even slightly increases when $\alpha = 1.2$ and $d$ grows from 70 to 100.

**Neural networks.** In our second set of experiments, we consider the real data setting used in [6]: a multi-layer fully connected neural network with ReLu activations on the MNIST dataset. We adapted the code provided in [6] and we provide our version in https://github.com/umutsimsekli/sgd_first_exit_time. For this model, we followed a similar methodology: we monitored the first exit time by

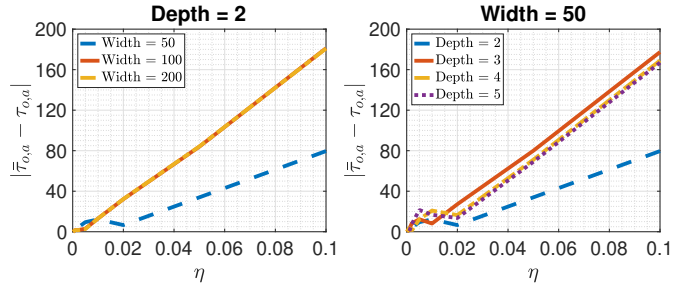

Figure 3: Results of the neural network experiments.

varying the $\eta$, the number of layers (depth), and the number of neurons per layer (width). Since a local minimum is not analytically available, we first trained the networks with SGD until a vicinity of a local minimum is reached with at least 90% accuracy, then we measured the first exit times with $a = 1$ and $\varepsilon = 0.1$. In order to have a prominent level of gradient noise, we set the mini-batch size $b = 10$ and we did not add explicit Gaussian or Lévy noise. The results are given in Figure 3. We observe that, even with pure gradient noise, the error in the exit time behaves very similarly to the one that we observed in Figure 2(a), hence supporting our theory. We further observe that, the error has a better dependency when the width and depth are relatively small, whereas the slope of the error increases for larger width and depth. This result shows that, to inherit the metastability properties of the continuous-time SDE, we need to use a smaller $\eta$ as we increase the size of the network. Note that this result does not conflict with Figure 2(d), since changing the width and depth does not simply change $d$, it also changes the landscape of the problem.

## 6   Conclusion

We studied SGD under a heavy-tailed gradient noise model, which has been empirically justified for a variety of deep learning tasks. While a continuous-time limit of SGD can be used as a proxy for investigating the metastability of SGD under this model, the system might behave differently once discretized. Addressing this issue, we derived explicit conditions for the step-size such that the discrete-time system can inherit the metastability behavior of its continuous-time limit. We illustrated our results on a synthetic model and neural networks. A natural next step four our study would be analyzing the generalization properties of SGD under such heavy-tailed gradient noise.

## Acknowledgments

We are grateful to Peter Tankov for providing us the derivations for the Girsanov-like change of measures. This work is partly supported by the French National Research Agency (ANR) as a part of the FBIMATRIX (ANR-16-CE23-0014) project, and by the industrial chair Data science & Artificial Intelligence from Télécom Paris. Mert Gürbüzbalaban acknowledges support from the grants NSF DMS-1723085 and NSF CCF-1814888.

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
