[Supplementary Material]

# First Exit Time Analysis of Stochastic Gradient Descent Under Heavy-Tailed Gradient Noise
## SUPPLEMENTARY DOCUMENT

**Thanh Huy Nguyen[1], Umut Şimşekli[1,2], Mert Gürbüzbalaban[3], Gaël Richard[1]**
1: LTCI, Télécom Paris, Institut Polytechnique de Paris, France
2: Department of Statistics, University of Oxford, UK
3: Dept. of Management Science and Information Systems, Rutgers Business School, NJ, USA

## S1 More details on Assumption A 6

In this section, we provide the precise expressions of the constants given in Assumption A 6.

For a given $\delta > 0$, $t = K\eta$, and for some $C > 0$, the step-size satisfies the following condition:

$$0 < \eta \le \min\left\{1, \frac{m}{M^2}, \left(\frac{\delta^2}{2K_1 t^2}\right)^{\frac{1}{\gamma^2+2\gamma-1}}, \left(\frac{\delta^2}{2K_2 t^2}\right)^{\frac{1}{2\gamma}}, \left(\frac{\delta^2}{2K_3 t^2}\right)^{\frac{\alpha}{2\gamma}}, \left(\frac{\delta^2}{2K_4 t^2}\right)^{\frac{1}{\gamma}}\right\},$$

where $\varepsilon$ is as in (7), the constants $m, M, b$ are defined by **A3**– **A5** and

$$K_1 \triangleq \frac{CM^{2+2\gamma}3^\gamma}{\varepsilon^2\sigma^2}\max\left\{(2(b+m))^{\gamma^2}, 2^{\gamma^2}B^{2\gamma^2}, d\varepsilon^{2\gamma^2}R_1, d\varepsilon^{2\gamma^2}R_2\right\},$$

$$K_2 \triangleq \frac{CM^{2+2\gamma}3^\gamma}{2\varepsilon^2\sigma^2}\left(\mathbb{E}\|W(0)\|^{2\gamma^2} + B^2/M^2\right),$$

$$K_3 \triangleq \frac{M^2 3^\gamma \varepsilon^{2\gamma-2} d^{2\gamma}}{2\sigma^2}\left(\frac{2^{2\gamma}\Gamma((1+2\gamma)/2)\Gamma(1-2\gamma/\alpha)}{\Gamma(1/2)\Gamma(1-\gamma)}\right),$$

$$K_4 \triangleq \frac{M^2 3^\gamma \varepsilon^{2\gamma-2} d^{2\gamma}}{2\sigma^2}\left(2^\gamma\Gamma(\frac{2\gamma+1}{2})/\sqrt{\pi}\right),$$

with

$$R_1 \triangleq \left(\frac{2^{2\gamma^2}\Gamma((1+2\gamma^2)/2)\Gamma(1-2\gamma^2/\alpha)}{\Gamma(1/2)\Gamma(1-\gamma^2)}\right), R_2 \triangleq \left(2^{\gamma^2}\Gamma\left(\frac{2\gamma^2+1}{2}\right)/\sqrt{\pi}\right).$$

## S2 Proof of Theorem 2

*Proof.* Note that $(W^1, \ldots, W^K) \in A$ is equivalent to $\bar{\tau}_{0,a}(\varepsilon) > K$. Hence, from Lemma S4, the remaining task is to upper-bound $\mathbb{P}[(W(\eta), \ldots, W(K\eta)) \in A]$:

$$\mathbb{P}[(W(\eta), \ldots, W(K\eta)) \in A] \le \mathbb{P}[(W(\eta), \ldots, W(K\eta)) \in A \cap B] + \mathbb{P}[(W(\eta), \ldots, W(K\eta)) \in B^c]$$
$$\le \mathbb{P}[\tau_{\xi,a}(\varepsilon) > K\eta] + \mathbb{P}[(W(\eta), \ldots, W(K\eta)) \in B^c],$$

and to lower-bound it:

$$\mathbb{P}[(W(\eta), \ldots, W(K\eta)) \in A] \ge \mathbb{P}[\tau_{-\xi,a}(\varepsilon) > K\eta] - \mathbb{P}[(W(\eta), \ldots, W(K\eta)) \in B^c].$$

By Lemma S1, the final result follows. $\square$

**Lemma S1.** *There exist constants $C$, $C_1$ and $C_\alpha$ such that:*

$$\mathbb{P}[(W(\eta), \ldots, W(K\eta)) \in B^c] \le \frac{C_1(K\eta(d\varepsilon+1)+1)^\gamma e^{M\eta}M\eta}{\xi} + 1 - \left(1 - Cde^{-\xi^2 e^{-2M\eta}(\varepsilon\sigma)^{-2}/(16d\eta)}\right)^K$$
$$+ 1 - \left(1 - C_\alpha d^{1+\alpha/2}\eta e^{\alpha M\eta}\varepsilon^\alpha\xi^{-\alpha}\right)^K,$$

*Proof.* We have for $t \in [k\eta, (k+1)\eta]$,

$$\|W(t) - W(k\eta)\| \leq \int_{k\eta}^{t} \|\nabla f(W(s))\| \mathrm{d}s + \varepsilon\sigma\|B(t) - B(k\eta)\| + \varepsilon\|L^{\alpha}(t) - L^{\alpha}(k\eta)\|$$

$$\leq \int_{k\eta}^{t} \|\nabla f(W(s)) - \nabla f(W(k\eta))\| \mathrm{d}s + \eta\|\nabla f(W(k\eta))\| + \varepsilon\sigma\|B(t) - B(k\eta)\|$$

$$+ \varepsilon\|L^{\alpha}(t) - L^{\alpha}(k\eta)\|$$

$$\leq \int_{k\eta}^{t} M\|W(s) - W(k\eta)\|^{\gamma} \mathrm{d}s + \eta(M\|W(k\eta)\|^{\gamma} + B) + \varepsilon\sigma\|B(t) - B(k\eta)\|$$

$$+ \varepsilon\|L^{\alpha}(t) - L^{\alpha}(k\eta)\|.$$

For $\gamma < 1$, using that $\|W(s) - W(k\eta)\|^{\gamma} \leq \|W(s) - W(k\eta)\| + 1$, we get:

$$\|W(t) - W(k\eta)\| \leq \int_{k\eta}^{t} M\|W(s) - W(k\eta)\| \mathrm{d}s + \eta(M\|W(k\eta)\|^{\gamma} + B + M)$$

$$+ \varepsilon\sigma \sup_{t \in [k\eta, (k+1)\eta]} \|B(t) - B(k\eta)\| + \varepsilon \sup_{t \in [k\eta, (k+1)\eta]} \|L^{\alpha}(t) - L^{\alpha}(k\eta)\|.$$

Then the Gronwall lemma gives:

$$\sup_{t \in [k\eta, (k+1)\eta]} \|W(t) - W(k\eta)\| \leq e^{M\eta} \Big[ \eta(M\|W(k\eta)\|^{\gamma} + B + M) + \varepsilon\sigma \sup_{t \in [k\eta, (k+1)\eta]} \|B(t) - B(k\eta)\|$$

$$+ \varepsilon \sup_{t \in [k\eta, (k+1)\eta]} \|L^{\alpha}(t) - L^{\alpha}(k\eta)\| \Big].$$

Hence,

$$\max_{0 \leq k \leq K-1} \sup_{t \in [k\eta, (k+1)\eta]} \|W(t) - W(k\eta)\| \leq e^{M\eta} \Big[ \eta(M \max_{0 \leq k \leq K-1} \|W(k\eta)\|^{\gamma} + B + M)$$

$$+ \varepsilon\sigma \max_{0 \leq k \leq K} \sup_{t \in [k\eta, (k+1)\eta]} \|B(t) - B(k\eta)\|$$

$$+ \varepsilon \max_{0 \leq k \leq K-1} \sup_{t \in [k\eta, (k+1)\eta]} \|L^{\alpha}(t) - L^{\alpha}(k\eta)\| \Big].$$

By Lemma 7.1 in [1], Lemma S4 in [2] and Markov's inequality, for any $u > 0$, we have:

$$\mathbb{P}[\max_{0 \leq k \leq K-1} \|W(k\eta)\|^{\gamma} \geq u] \leq \frac{\mathbb{E}[\max_{0 \leq k \leq K-1} \|W(k\eta)\|^{\gamma}]}{u} \leq \frac{C_1(K\eta(d\varepsilon + 1) + 1)^{\gamma}}{u},$$

where $C_1$ is a constant independent of $K, \eta, \varepsilon$ and $d$. By Lemma S3, we have:

$$\mathbb{P}[\max_{k \in [0,...,K-1]} \sup_{t \in [k\eta, (k+1)\eta]} \|B(t) - B(k\eta)\| \geq u] \leq 1 - \Big(1 - Cde^{-u^2/(d\eta)}\Big)^K$$

and

$$\mathbb{P}[\max_{k \in [0,...,K-1]} \sup_{t \in [k\eta, (k+1)\eta]} \|L^{\alpha}(t) - L^{\alpha}(k\eta)\| \geq u] \leq 1 - \Big(1 - C_{\alpha}d^{1+\alpha/2}\eta u^{-\alpha}\Big)^K.$$

Finally, we get:

$$\mathbb{P}[(W(\eta),\dots,W(K\eta)) \in B^c] \leq \mathbb{P}[\max_{0\leq k\leq K-1} \sup_{t\in[k\eta,(k+1)\eta]} \|W(t) - W(k\eta)\| > \xi]$$

$$\leq \mathbb{P}[e^{M\eta}\eta M \max_{0\leq k\leq K-1} \|W(k\eta)\|^\gamma \geq \xi/4]$$

$$+ \mathbb{P}[e^{M\eta}\eta(B+M) \geq \xi/4]$$

$$+ \mathbb{P}[e^{M\eta} \max_{k\in[0,\dots,K-1]} \sup_{t\in[k\eta,(k+1)\eta]} \|B(t)-B(k\eta)\| \geq (\varepsilon\sigma)^{-1}\xi/4]$$

$$+ \mathbb{P}[e^{M\eta} \max_{k\in[0,\dots,K-1]} \sup_{t\in[k\eta,(k+1)\eta]} \|L^\alpha(t)-L^\alpha(k\eta)\| \geq \varepsilon^{-1}\xi/4]$$

$$\leq \frac{C_1(K\eta(d\varepsilon+1)+1)^\gamma e^{M\eta}M\eta}{\xi} + 1 - \left(1 - Cde^{-\xi^2 e^{-2M\eta}(\varepsilon\sigma)^{-2}/(16d\eta)}\right)^K$$

$$+ 1 - \left(1 - C_\alpha d^{1+\alpha/2}\eta e^{\alpha M\eta}\varepsilon^\alpha \xi^{-\alpha}\right)^K.$$

$\square$

Now we prove the following lemma.

**Lemma S2.** *There exist constants $C$ and $C_\alpha$ such that:*

$$\max_{k\in[0,\dots,K-1]} \mathbb{P}[\sup_{t\in[k\eta,(k+1)\eta]} \|B(t)-B(k\eta)\| \geq u] \leq Cde^{-cu^2/(d\eta)}.$$

$$\max_{k\in[0,\dots,K-1]} \mathbb{P}[\sup_{t\in[k\eta,(k+1)\eta]} \|L^\alpha(t)-L^\alpha(k\eta)\| \geq u] \leq C_\alpha d^{1+\alpha/2}\eta u^{-\alpha}.$$

*Proof.* To prove the results, we begin with the known results for Brownian motion and $\alpha$-stable Lévy motion:

$$\mathbb{P}[|[B(1)]_i| \geq u] \leq Ce^{-u^2},$$
$$\mathbb{P}[|[L^\alpha(1)]_i| \geq u] \leq C_\alpha u^{-\alpha},$$

where $C$ and $C_\alpha$ are positive constants, $[B(1)]_i$ and $[L^\alpha(1)]_i$ denote the $i$-th component of the motions respectively, for $i$ from 1 to $d$. By reflection principle for Brownian motion and $\alpha$-stable Lévy motion, we have

$$\mathbb{P}[\sup_{t\in[k\eta,(k+1)\eta]} |[B(t)-B(k\eta)]_i| \geq u] \leq 2\mathbb{P}[|[B(\eta)]_i| \geq u] = 2\mathbb{P}[|[B(1)]_i| \geq u/\eta^{1/2}],$$

$$\mathbb{P}[\sup_{t\in[k\eta,(k+1)\eta]} |[L^\alpha(t)-L^\alpha(k\eta)]_i| \geq u] \leq 2\mathbb{P}[|[L^\alpha(\eta)]_i| \geq u] = 2\mathbb{P}[|[L^\alpha(1)]_i| \geq u/\eta^{1/\alpha}].$$

Since $\sup_{t\in[k\eta,(k+1)\eta]} \|B(t)-B(k\eta)\|^2 \leq \sum_{i=1}^d \sup_{t\in[k\eta,(k+1)\eta]} |[B(t)-B(k\eta)]_i|^2$, we have

$$\mathbb{P}[\sup_{t\in[k\eta,(k+1)\eta]} \|B(t)-B(k\eta)\| \geq u] = \mathbb{P}[\sup_{t\in[k\eta,(k+1)\eta]} \|B(t)-B(k\eta)\|^2 \geq u^2]$$

$$\leq \sum_{i=1}^d \mathbb{P}[\sup_{t\in[k\eta,(k+1)\eta]} |[B(t)-B(k\eta)]_i|^2 \geq u^2/d]$$

$$\leq \sum_{i=1}^d 2\mathbb{P}[|[B(1)]_i| \geq u/(d\eta)^{1/2}]$$

$$\leq 2Cde^{-u^2/(d\eta)}.$$

Similarly, we have

$$\mathbb{P}[\sup_{t\in[k\eta,(k+1)\eta]} \|L^\alpha(t)-L^\alpha(k\eta)\| \geq u] \leq \sum_{i=1}^d 2\mathbb{P}[|[L^\alpha(1)]_i| \geq u/(d^{1/2}\eta^{1/\alpha})]$$

$$\leq 2C_\alpha d^{1+\alpha/2}\eta u^{-\alpha}.$$

The constants $C$ and $C_\alpha$ do not depend on $k$, hence we have the conclusion. $\square$

**Lemma S3.** *The following estimates hold:*

$$\mathbb{P}\big[\max_{k\in[0,...,K-1]}\sup_{t\in[k\eta,(k+1)\eta]}\|B(t)-B(k\eta)\|\geq u\big]\leq 1-\Big(1-Cde^{-u^2/(d\eta)}\Big)^K$$

$$\mathbb{P}\big[\max_{k\in[0,...,K-1]}\sup_{t\in[k\eta,(k+1)\eta]}\|L^\alpha(t)-L^\alpha(k\eta)\|\geq u\big]\leq 1-\Big(1-C_\alpha d^{1+\alpha/2}\eta u^{-\alpha}\Big)^K.$$

*Proof.* We have

$$\mathbb{P}\big[\max_{k\in[0,...,K-1]}\sup_{t\in[k\eta,(k+1)\eta]}\|B(t)-B(k\eta)\|\geq u\big]=1-\mathbb{P}\big[\max_{k\in[0,...,K-1]}\sup_{t\in[k\eta,(k+1)\eta]}\|B(t)-B(k\eta)\|<u\big]$$

$$=1-\prod_{k=0}^{K-1}\mathbb{P}\big[\sup_{t\in[k\eta,(k+1)\eta]}\|B(t)-B(k\eta)\|<u\big]$$

$$=1-\prod_{k=0}^{K-1}\Big(1-\mathbb{P}\big[\sup_{t\in[k\eta,(k+1)\eta]}\|B(t)-B(k\eta)\|\geq u\big]\Big)$$

$$\leq 1-\prod_{k=0}^{K-1}\Big(1-Cde^{-u^2/(d\eta)}\Big)$$

$$=1-\Big(1-Cde^{-u^2/(d\eta)}\Big)^K.$$

Similarly, we have

$$\mathbb{P}\big[\max_{k\in[0,...,K-1]}\sup_{t\in[k\eta,(k+1)\eta]}\|L^\alpha(t)-L^\alpha(k\eta)\|\geq u\big]\leq 1-\Big(1-C_\alpha d^{1+\alpha/2}\eta u^{-\alpha}\Big)^K.$$

$\square$

**Lemma S4.** *Suppose that assumptions A3 and A4 hold. Then, for any $\delta>0$, we have:*

$$\mathbb{P}[(W(\eta),\ldots,W(K\eta))\in A]-\delta\leq\mathbb{P}[(\hat{W}(\eta),\ldots,\hat{W}(K\eta))\in A]\leq\mathbb{P}[(W(\eta),\ldots,W(K\eta))\in A]+\delta,$$

*provided that*

$$0<\eta\leq\min\Big\{1,\frac{m}{M^2},\Big(\frac{\delta^2}{2K_1t^2}\Big)^{\frac{1}{\gamma^2+2\gamma-1}},\Big(\frac{\delta^2}{2K_2t^2}\Big)^{\frac{1}{2\gamma}},\Big(\frac{\delta^2}{2K_3t^2}\Big)^{\frac{\alpha}{2\gamma}},\Big(\frac{\delta^2}{2K_4t^2}\Big)^{\frac{1}{\gamma}}\Big\},$$

*Proof.* By optimal coupling between two probability measure ([3], Theorem 5.2), there exists a coupling $\mathbf{M}$ of $(W(s))_{0\leq s\leq K\eta}$ and $(\hat{W}(s))_{0\leq s\leq K\eta}$ such that

$$\mathbb{P}_{\mathbf{M}}[(W(s))_{0\leq s\leq K\eta}\neq(\hat{W}(s))_{0\leq s\leq K\eta}]=\|\mu_{K\eta}-\hat{\mu}_{K\eta}\|_{TV},$$

where $TV$ denotes the total variation distance. By Pinsker's inequality, we also have

$$\|\mu_{K\eta}-\hat{\mu}_{K\eta}\|_{TV}^2\leq\frac{1}{2}\mathrm{KL}(\hat{\mu}_{K\eta},\mu_{K\eta}).$$

Then,

$$\mathbb{P}_{\mathbf{M}}[(W(\eta),\ldots,W(K\eta))\neq(\hat{W}(\eta),\ldots,\hat{W}(K\eta))]\leq\mathbb{P}_{\mathbf{M}}[(W(s))_{0\leq s\leq K\eta}\neq(\hat{W}(s))_{0\leq s\leq K\eta}]$$

$$\leq\Big(\frac{1}{2}\mathrm{KL}(\hat{\mu}_{K\eta},\mu_{K\eta})\Big)^{1/2}.$$

From the following inequalities

$$\mathbb{P}_{\mathbf{M}}[(W(\eta),\ldots,W(K\eta))\in A]-\mathbb{P}_{\mathbf{M}}[(W(\eta),\ldots,W(K\eta))\neq(\hat{W}(\eta),\ldots,\hat{W}(K\eta))]\leq\mathbb{P}_{\mathbf{M}}[(\hat{W}(\eta),\ldots,\hat{W}(K\eta))\in A]$$

$$\mathbb{P}_{\mathbf{M}}[(\hat{W}(\eta),\ldots,\hat{W}(K\eta))\in A]\leq\mathbb{P}_{\mathbf{M}}[(W(\eta),\ldots,W(K\eta))\in A]+\mathbb{P}_{\mathbf{M}}[(W(\eta),\ldots,W(K\eta))\neq(\hat{W}(\eta),\ldots,\hat{W}(K\eta))],$$

we arrive at

$$\mathbb{P}[(W(\eta),\ldots,W(K\eta))\in A]-\Big(\frac{1}{2}\mathrm{KL}(\hat{\mu}_{K\eta},\mu_{K\eta})\Big)^{1/2}\leq\mathbb{P}[(\hat{W}(\eta),\ldots,\hat{W}(K\eta))\in A]$$

$$\mathbb{P}[(\hat{W}(\eta),\ldots,\hat{W}(K\eta))\in A]\leq\mathbb{P}[(W(\eta),\ldots,W(K\eta))\in A]+\Big(\frac{1}{2}\mathrm{KL}(\hat{\mu}_{K\eta},\mu_{K\eta})\Big)^{1/2}.$$

By Theorem 3, we have the desired inequalities. $\square$

## S3 Proof of Theorem 3

### S3.1 A Girsanov-Type Change of Measures

In this section we will derive a Girsanov-type change of measure [4, 5] for the SDE considered in (6). Let $\mathbb{P}$ denote the law of $W(t)$ and $\mathbb{Q}$ be an equivalent measure defined by

$$\frac{d\mathbb{Q}}{d\mathbb{P}}\Big|_{\mathcal{F}_T} = \exp\left(\int_0^T \phi_t dB_t - \frac{1}{2}\int_0^T \phi_t^2 dt\right), \tag{S1}$$

where $\mathcal{F}_T$ denotes the filtration upto time $T$. Then the process $B^\phi$ defined by $B^\phi(t) = B(t) - \int_0^t \phi_s ds$ is a $\mathbb{Q}$-Brownian motion. With the choice of $\phi_t$ given in **A2**, we see that $W$ satisfies $dW(t) = b(W)dt + \varepsilon\sigma dB^\phi(t) + \varepsilon dL^\alpha(t)$. Since this equation has a unique solution (constructed explicitly with the Euler scheme), we conclude that $W$ has the same law under $\mathbb{Q}$ as $\hat{W}$ under $\mathbb{P}$.

We thus have:

$$\mathrm{KL}(\hat{\mu}_t, \mu_t) = \mathrm{KL}(\mathbb{P}_t, \mathbb{Q}_t) = \mathbb{E}^{\mathbb{P}}\left[\log\frac{d\mathbb{P}}{d\mathbb{Q}}\Big|_{\mathcal{F}_t}\right] = \frac{1}{2\varepsilon^2\sigma^2}\mathbb{E}^{\mathbb{P}}\left[\int_0^t \|b(\hat{W}) + \nabla f(\hat{W}(s))\|^2 ds\right] \tag{S2}$$

By using the same steps of the proof of [6][Lemma 3.6], we obtain

$$\mathrm{KL}(\hat{\mu}_t, \mu_t) = \frac{1}{2\varepsilon^2\sigma^2}\sum_{j=0}^{k-1}\int_{j\eta}^{(j+1)\eta}\mathbb{E}\|\nabla f(\hat{W}(s)) - \nabla f(\hat{W}(j\eta))\|^2 ds \tag{S3}$$

$$\leq \frac{M^2}{2\varepsilon^2\sigma^2}\sum_{j=0}^{k-1}\int_{j\eta}^{(j+1)\eta}\mathbb{E}\|\hat{W}(s) - \hat{W}(j\eta)\|^{2\gamma} ds. \tag{S4}$$

### S3.2 KL Bound for the Discretized Process

**Theorem S1.** *Under assumptions A3 and A4 we have, for $0 < \eta \leq \min\{1, \frac{m}{M^2}\}$,*

$$\begin{aligned}
\mathrm{KL}(\hat{\mu}_t, \mu_t) \leq &\frac{M^2 3^\gamma}{2\varepsilon^2\sigma^2}k\eta\Bigg(CM^{2\gamma}\eta^{2\gamma}\Big(\mathbb{E}\|\hat{W}(0)\|^{2\gamma^2} \\
&+ \frac{k-1}{2}\Big((2\eta(b+m))^{\gamma^2} + 2^{\gamma^2}(\eta B)^{2\gamma^2} + \varepsilon^{2\gamma^2}\eta^{\frac{2\gamma^2}{\alpha}}d\Big(\frac{2^{2\gamma^2}\Gamma((1+2\gamma^2)/2)\Gamma(1-2\gamma^2/\alpha)}{\Gamma(1/2)\Gamma(1-\gamma^2)}\Big) \\
&+ \varepsilon^{2\gamma^2}\eta^{\gamma^2}d\Big(2^{\gamma^2}\frac{\Gamma\left(\frac{2\gamma^2+1}{2}\right)}{\sqrt{\pi}}\Big)\Big) + \frac{B^2}{M^2}\Big) + (\varepsilon\eta^{1/\alpha})^{2\gamma}d^{2\gamma}\Big(\frac{2^{2\gamma}\Gamma((1+2\gamma)/2)\Gamma(1-2\gamma/\alpha)}{\Gamma(1/2)\Gamma(1-\gamma)}\Big) \\
&+ (\varepsilon\eta^{1/2})^{2\gamma}d^{2\gamma}\Big(2^\gamma\frac{\Gamma\left(\frac{2\gamma+1}{2}\right)}{\sqrt{\pi}}\Big)\Bigg) \\
\leq &K_1 k^2\eta^{1+2\gamma+\gamma^2} + K_2 k\eta^{1+2\gamma} + K_3 k\eta^{1+\frac{2\gamma}{\alpha}} + K_4 k\eta^{1+\gamma},
\end{aligned}$$

*where*

$$K_1 \triangleq \frac{CM^{2+2\gamma}3^\gamma}{\varepsilon^2\sigma^2}\max\Big\{(2(b+m))^{\gamma^2}, 2^{\gamma^2}B^{2\gamma^2}, \varepsilon^{2\gamma^2}d\Big(\frac{2^{2\gamma^2}\Gamma((1+2\gamma^2)/2)\Gamma(1-2\gamma^2/\alpha)}{\Gamma(1/2)\Gamma(1-\gamma^2)}\Big),$$

$$\varepsilon^{2\gamma^2}d\Big(2^{\gamma^2}\frac{\Gamma\left(\frac{2\gamma^2+1}{2}\right)}{\sqrt{\pi}}\Big)\Big)\Big\},$$

$$K_2 \triangleq \frac{CM^{2+2\gamma}3^\gamma}{2\varepsilon^2\sigma^2}\Big(\mathbb{E}\|\hat{W}(0)\|^{2\gamma^2} + \frac{B^2}{M^2}\Big),$$

$$K_3 \triangleq \frac{M^2 3^\gamma \varepsilon^{2\gamma-2}d^{2\gamma}}{2\sigma^2}\Big(\frac{2^{2\gamma}\Gamma((1+2\gamma)/2)\Gamma(1-2\gamma/\alpha)}{\Gamma(1/2)\Gamma(1-\gamma)}\Big),$$

$$K_4 \triangleq \frac{M^2 3^\gamma \varepsilon^{2\gamma-2}d^{2\gamma}}{2\sigma^2}\Big(2^\gamma\frac{\Gamma\left(\frac{2\gamma+1}{2}\right)}{\sqrt{\pi}}\Big).$$

*Proof.* Let us consider the term $\hat{W}(s) - \hat{W}(j\eta)$, for $s \in [j\eta, (j+1)\eta]$:

$$\hat{W}(s) - \hat{W}(j\eta) = -(s - j\eta)\nabla f(\hat{W}(j\eta)) + \varepsilon(L_s - L_{j\eta}) + \varepsilon(B_s - B_{j\eta}) \tag{S5}$$

$$\triangleq T_1 + T_2 + T_3 \tag{S6}$$

Using this equation and (S4), we obtain:

$$\mathrm{KL}(\hat{\mu}_t, \mu_t) \leq \frac{M^2}{2\varepsilon^2\sigma^2} \sum_{j=0}^{k-1} \int_{j\eta}^{(j+1)\eta} \mathbb{E}\|T_1 + T_2 + T_3\|^{2\gamma}\, \mathrm{d}s \tag{S7}$$

$$\leq \frac{M^2}{2\varepsilon^2\sigma^2} \sum_{j=0}^{k-1} \int_{j\eta}^{(j+1)\eta} \mathbb{E}\Big(\|T_1 + T_2 + T_3\|^2\Big)^\gamma \mathrm{d}s \tag{S8}$$

$$\leq \frac{M^2}{2\varepsilon^2\sigma^2} \sum_{j=0}^{k-1} \int_{j\eta}^{(j+1)\eta} \mathbb{E}\Big(3\|T_1\|^2 + 3\|T_2\|^2 + 3\|T_3\|^2\Big)^\gamma \mathrm{d}s \tag{S9}$$

$$\leq \frac{M^2 3^\gamma}{2\varepsilon^2\sigma^2} \sum_{j=0}^{k-1} \int_{j\eta}^{(j+1)\eta} \mathbb{E}\Big(\|T_1\|^{2\gamma} + \|T_2\|^{2\gamma} + \|T_3\|^{2\gamma}\Big) \mathrm{d}s \tag{S10}$$

where (S9) is obtained from $(a+b)^\gamma \leq a^\gamma + b^\gamma$ since $\gamma \in (0,1)$ and $a, b \geq 0$.

Since $2\gamma > 1$, we have by Lemma S6

$$\mathbb{E}\|T_2\|^{2\gamma} = \mathbb{E}\|\varepsilon(s - j\eta)^{1/\alpha}L^\alpha(1))\|^{2\gamma}$$

$$\leq (\varepsilon\eta^{1/\alpha})^{2\gamma}\mathbb{E}\|L^\alpha(1)\|^{2\gamma}$$

$$\leq (\varepsilon\eta^{1/\alpha})^{2\gamma} d^{2\gamma}\Big(\frac{2^{2\gamma}\Gamma((1+2\gamma)/2)\Gamma(1-2\gamma/\alpha)}{\Gamma(1/2)\Gamma(1-\gamma)}\Big),$$

and by Corollary S1,

$$\mathbb{E}\|T_3\|^{2\gamma} = \mathbb{E}\|\varepsilon(s - j\eta)^{1/2}B(1))\|^{2\gamma}$$

$$\leq (\varepsilon\eta^{1/2})^{2\gamma}\mathbb{E}\|B(1)\|^{2\gamma}$$

$$\leq (\varepsilon\eta^{1/2})^{2\gamma} d^{2\gamma}\left(2^\gamma\frac{\Gamma\left(\frac{2\gamma+1}{2}\right)}{\sqrt{\pi}}\right),$$

By definition, we have

$$\mathbb{E}\|T_1\|^{2\gamma} = \mathbb{E}\|(s - j\eta)\nabla f(\hat{W}(j\eta))\|^{2\gamma} \tag{S11}$$

$$\leq \eta^{2\gamma}\mathbb{E}\|\nabla f(\hat{W}(j\eta))\|^{2\gamma} \tag{S12}$$

$$\leq \eta^{2\gamma}\mathbb{E}(M\|\hat{W}(j\eta)\|^\gamma + B)^{2\gamma} \tag{S13}$$

$$\leq CM^{2\gamma}\eta^{2\gamma}\mathbb{E}\left(\|\hat{W}(j\eta)\|_\gamma^\gamma + \Big(\frac{B^{1/\gamma}}{M^{1/\gamma}}\Big)^\gamma\right)^{2\gamma} \tag{S14}$$

$$\leq CM^{2\gamma}\eta^{2\gamma}\mathbb{E}\left(\|\hat{W}'(j\eta)\|_\gamma^\gamma\right)^{2\gamma} \tag{S15}$$

where we used the equivalence of $\ell_p$-norms and $\hat{W}'(j\eta)$ is the concatenation of $\hat{W}(j\eta)$ and $\frac{B^{1/\gamma}}{M^{1/\gamma}}$. We then obtain

$$\mathbb{E}\|T_1\|^{2\gamma} \leq CM^{2\gamma}\eta^{2\gamma}\mathbb{E}\|\hat{W}'(j\eta)\|_\gamma^{2\gamma^2} \tag{S16}$$

$$\leq CM^{2\gamma}\eta^{2\gamma}\mathbb{E}\|\hat{W}'(j\eta)\|_{2\gamma^2}^{2\gamma^2} \tag{S17}$$

$$= CM^{2\gamma}\eta^{2\gamma}\mathbb{E}\Big(\|\hat{W}(j\eta)\|_{2\gamma^2}^{2\gamma^2} + \frac{B^2}{M^2}\Big) \tag{S18}$$

$$\leq CM^{2\gamma}\eta^{2\gamma}\Big(\mathbb{E}\|\hat{W}(j\eta)\|^{2\gamma^2} + \frac{B^2}{M^2}\Big). \tag{S19}$$

By combining the above inequalities and Lemma S8, we obtain

$$
\begin{aligned}
\mathrm{KL}(\hat{\mu}_t, \mu_t) \leq & \frac{M^2 3^\gamma}{2\varepsilon^2 \sigma^2} \sum_{j=0}^{k-1} \int_{j\eta}^{(j+1)\eta} \mathbb{E}\Big( \|T_1\|^{2\gamma} + \|T_2\|^{2\gamma} + \|T_3\|^{2\gamma} \Big) \, \mathrm{d}s \\
\leq & \frac{M^2 3^\gamma}{2\varepsilon^2 \sigma^2} \sum_{j=0}^{k-1} \int_{j\eta}^{(j+1)\eta} \Bigg( CM^{2\gamma}\eta^{2\gamma} \Big( \mathbb{E}\|\hat{W}(0)\|^{2\gamma^2} \\
& + j\Big( (2\eta(b+m))^{\gamma^2} + 2^{\gamma^2}(\eta B)^{2\gamma^2} + \varepsilon^{2\gamma^2}\eta^{\frac{2\gamma^2}{\alpha}} d\Big( \frac{2^{2\gamma^2}\Gamma((1+2\gamma^2)/2)\Gamma(1-2\gamma^2/\alpha)}{\Gamma(1/2)\Gamma(1-\gamma^2)} \Big) \\
& + \varepsilon^{2\gamma^2}\eta^{\gamma^2} d\Big( 2^{\gamma^2} \frac{\Gamma\left(\frac{2\gamma^2+1}{2}\right)}{\sqrt{\pi}} \Big) \Big) + \frac{B^2}{M^2} \Big) + (\varepsilon\eta^{1/\alpha})^{2\gamma} d^{2\gamma} \Big( \frac{2^{2\gamma}\Gamma((1+2\gamma)/2)\Gamma(1-2\gamma/\alpha)}{\Gamma(1/2)\Gamma(1-\gamma)} \Big) \\
& + (\varepsilon\eta^{1/2})^{2\gamma} d^{2\gamma} \Big( 2^\gamma \frac{\Gamma\left(\frac{2\gamma+1}{2}\right)}{\sqrt{\pi}} \Big) \Bigg) \, \mathrm{d}s \\
= & \frac{M^2 3^\gamma}{2\varepsilon^2 \sigma^2} k\eta \Bigg( CM^{2\gamma}\eta^{2\gamma} \Big( \mathbb{E}\|\hat{W}(0)\|^{2\gamma^2} \\
& + \frac{k-1}{2}\Big( (2\eta(b+m))^{\gamma^2} + 2^{\gamma^2}(\eta B)^{2\gamma^2} + \varepsilon^{2\gamma^2}\eta^{\frac{2\gamma^2}{\alpha}} d\Big( \frac{2^{2\gamma^2}\Gamma((1+2\gamma^2)/2)\Gamma(1-2\gamma^2/\alpha)}{\Gamma(1/2)\Gamma(1-\gamma^2)} \Big) \\
& + \varepsilon^{2\gamma^2}\eta^{\gamma^2} d\Big( 2^{\gamma^2} \frac{\Gamma\left(\frac{2\gamma^2+1}{2}\right)}{\sqrt{\pi}} \Big) \Big) + \frac{B^2}{M^2} \Big) + (\varepsilon\eta^{1/\alpha})^{2\gamma} d^{2\gamma} \Big( \frac{2^{2\gamma}\Gamma((1+2\gamma)/2)\Gamma(1-2\gamma/\alpha)}{\Gamma(1/2)\Gamma(1-\gamma)} \Big) \\
& + (\varepsilon\eta^{1/2})^{2\gamma} d^{2\gamma} \Big( 2^\gamma \frac{\Gamma\left(\frac{2\gamma+1}{2}\right)}{\sqrt{\pi}} \Big) \Bigg).
\end{aligned}
$$

By defining the constants $K_1, K_2, K_3$ and $K_4$ as in the statement of the Theorem, we directly have the conclusion. $\quad\square$

### S3.3  Proof of Theorem 3

*Proof.* By Theorem S1, we have

$$
\mathrm{KL}(\hat{\mu}_t, \mu_t) \leq K_1 k^2 \eta^{1+2\gamma+\gamma^2} + K_2 k\eta^{1+2\gamma} + K_3 k\eta^{1+\frac{2\gamma}{\alpha}} + K_4 k\eta^{1+\gamma}.
$$

We can easily check that, for example, if $0 < \eta \leq \left( \frac{\delta^2}{2K_1 t^2} \right)^{\frac{1}{\gamma^2+2\gamma-1}}$, then $K_1 k^2 \eta^{1+2\gamma+\gamma^2} \leq \frac{\delta^2}{2}$. By the same arguments, we finally have

$$
\begin{aligned}
\mathrm{KL}(\hat{\mu}_t, \mu_t) \leq & \frac{\delta^2}{2} + \frac{\delta^2}{2} + \frac{\delta^2}{2} + \frac{\delta^2}{2} \\
= & 2\delta^2.
\end{aligned}
$$

This finalizes the proof. $\quad\square$

## S4  Technical Results

**Lemma S5.** *Under assumptions A3 and A4 we have*

$$
\|\nabla f(w)\| \leq M\|w\|^\gamma + B, \quad \forall w \in \mathbb{R}^d.
$$

*Proof.* By assumption **A3** we have

$$
\|\nabla f(w) - \nabla f(0)\| \leq M\|w - 0\|^\gamma.
$$

Since $\|\nabla f(0)\| \leq B$ by assumption **A4**, the conclusion follows. $\quad\square$

The next lemma is the result on the moments of the noise $L^\alpha(1)$.

**Lemma S6.** *The quantity $\mathbb{E}\|L^\alpha(1)\|^\lambda$ is finite for $0 \le \lambda < \alpha$. For details, we have*

*(a) If $1 < \lambda < \alpha$, then*

$$\mathbb{E}\|L^\alpha(1)\|^\lambda \le d^\lambda\Big(\frac{2^\lambda\Gamma((1+\lambda)/2)\Gamma(1-\lambda/\alpha)}{\Gamma(1/2)\Gamma(1-\lambda/2)}\Big).$$

*(b) If $0 \le \lambda \le 1$, then*

$$\mathbb{E}\|L^\alpha(1)\|^\lambda \le d\Big(\frac{2^\lambda\Gamma((1+\lambda)/2)\Gamma(1-\lambda/\alpha)}{\Gamma(1/2)\Gamma(1-\lambda/2)}\Big).$$

*Proof.* This is exactly Corollary S3 in [2]. □

For the moments of the noise $B(1)$, we first have the following lemma.

**Lemma S7.** *Let $X$ be a scalar standard Gaussian random variable. Then, for $\lambda > -1$, we have*

$$\mathbb{E}(|X|^\lambda) = 2^{\lambda/2}\frac{\Gamma\left(\frac{\lambda+1}{2}\right)}{\sqrt{\pi}},$$

*where $\Gamma$ denotes the Gamma function.*

*Proof.* The result is a direct consequence of equation (17) in [7]. □

**Corollary S1.** *The quantity $\mathbb{E}\|B(1)\|^\lambda$ is finite for $\lambda > -1$. For details, we have*

*(a) If $1 < \lambda < \alpha$, then*

$$\mathbb{E}\|B(1)\|^\lambda \le d^\lambda\left(2^{\lambda/2}\frac{\Gamma\left(\frac{\lambda+1}{2}\right)}{\sqrt{\pi}}\right).$$

*(b) If $0 \le \lambda \le 1$, then*

$$\mathbb{E}\|B(1)\|^\lambda \le d\left(2^{\lambda/2}\frac{\Gamma\left(\frac{\lambda+1}{2}\right)}{\sqrt{\pi}}\right).$$

*Proof.* Since $B(1)$, by definition, is a d-dimensional vector whose components are i.i.d standard Gaussian random variable $B_i(1)$ for $i \in \{1,\dots,d\}$, we have

$$\|B(1)\| \le \sum_{i=1}^{d}|B_i(1)|$$

(a) $1 < \lambda < \alpha$. By using Minkowski's inequality and Lemma S7,

$$(\mathbb{E}\|B(1)\|^\lambda)^{1/\lambda} \le \left(\mathbb{E}\Big[\Big(\sum_{i=1}^{d}|B_i(1)|\Big)^\lambda\Big]\right)^{1/\lambda}$$

$$\le \sum_{i=1}^{d}(\mathbb{E}|B_i(1)|^\lambda)^{1/\lambda}$$

$$= d\left(2^{\lambda/2}\frac{\Gamma\left(\frac{\lambda+1}{2}\right)}{\sqrt{\pi}}\right)^{1/\lambda}.$$

Thus, we have

$$\mathbb{E}\|B(1)\|^\lambda \le d^\lambda\left(2^{\lambda/2}\frac{\Gamma\left(\frac{\lambda+1}{2}\right)}{\sqrt{\pi}}\right).$$

(b) $0 \leq \lambda \leq 1$.

$$\mathbb{E}\|B(1)\|^{\lambda} \leq \mathbb{E}\Big[\Big(\sum_{i=1}^{d}|B_i(1)|\Big)^{\lambda}\Big]$$

$$\leq \sum_{i=1}^{d}\mathbb{E}|B_i(1)|^{\lambda}$$

$$= d\left(2^{\lambda/2}\frac{\Gamma\left(\frac{\lambda+1}{2}\right)}{\sqrt{\pi}}\right).$$

$\square$

**Lemma S8.** *For $0 < \eta \leq \frac{m}{M^2}$ and $s \in [j\eta, (j+1)\eta)$, we have the following estimates:*

*(a) If $1 < \lambda < \alpha$ then*

$$\mathbb{E}\|\hat{W}(j\eta)\|^{\lambda} \leq \Bigg(\Big(\big(\mathbb{E}\|\hat{W}(0)\|^{\lambda}\big)^{\frac{1}{\lambda}} + j\Big((2\eta(b+m))^{\frac{1}{2}} + 2^{\frac{1}{2}}\eta B + \varepsilon\eta^{\frac{1}{\alpha}}d\Big(\frac{2^{\lambda}\Gamma((1+\lambda)/2)\Gamma(1-\lambda/\alpha)}{\Gamma(1/2)\Gamma(1-\lambda/2)}\Big)^{\frac{1}{\lambda}}$$

$$+ \varepsilon\eta^{\frac{1}{2}}d\Big(2^{\lambda/2}\frac{\Gamma\left(\frac{\lambda+1}{2}\right)}{\sqrt{\pi}}\Big)^{\frac{1}{\lambda}}\Big)\Bigg)^{\lambda}.$$

*(b) If $0 \leq \lambda \leq 1$ then*

$$\mathbb{E}\|\hat{W}(j\eta)\|^{\lambda} \leq \mathbb{E}\|\hat{W}(0)\|^{\lambda} + j\Big((2\eta(b+m))^{\frac{\lambda}{2}} + 2^{\frac{\lambda}{2}}(\eta B)^{\lambda} + \varepsilon^{\lambda}\eta^{\frac{\lambda}{\alpha}}d\Big(\frac{2^{\lambda}\Gamma((1+\lambda)/2)\Gamma(1-\lambda/\alpha)}{\Gamma(1/2)\Gamma(1-\lambda/2)}\Big)$$

$$+ \varepsilon^{\lambda}\eta^{\frac{\lambda}{2}}d\Big(2^{\lambda/2}\frac{\Gamma\left(\frac{\lambda+1}{2}\right)}{\sqrt{\pi}}\Big)\Big).$$

*Proof.* The proof technique is similar to [2]. Let us denote the value $\mathbb{E}\|L^{\alpha}(1)\|^{\lambda}$ by $l_{\alpha,\lambda,d} < \infty$ and the value $\mathbb{E}\|B(1)\|^{\lambda}$ by $b_{\lambda,d} < \infty$. Starting from

$$\hat{W}((j+1)\eta) = \hat{W}(j\eta) - \eta\nabla f(\hat{W}(j\eta)) + \varepsilon\eta^{\frac{1}{\alpha}}L^{\alpha}(1) + \varepsilon\eta^{\frac{1}{2}}B(1),$$

we have either, by Minkowski, for $\lambda > 1$,

$$\Big(\mathbb{E}\|\hat{W}((j+1)\eta)\|^{\lambda}\Big)^{\frac{1}{\lambda}} \leq \Big(\mathbb{E}\|\hat{W}(j\eta) - \eta\nabla f(\hat{W}(j\eta))\|^{\lambda}\Big)^{\frac{1}{\lambda}} + \varepsilon\eta^{\frac{1}{\alpha}}\Big(\mathbb{E}\|L^{\alpha}(1)\|^{\lambda}\Big)^{\frac{1}{\lambda}} + \varepsilon\eta^{\frac{1}{2}}\Big(\mathbb{E}\|B(1)\|^{\lambda}\Big)^{\frac{1}{\lambda}}, \quad \text{(S20)}$$

or for $0 \leq \lambda \leq 1$),

$$\mathbb{E}\|\hat{W}((j+1)\eta)\|^{\lambda} \leq \mathbb{E}\|\hat{W}(j\eta) - \eta\nabla f(\hat{W}(j\eta))\|^{\lambda} + \varepsilon^{\lambda}\eta^{\frac{\lambda}{\alpha}}\mathbb{E}\|L^{\alpha}(1)\|^{\lambda} + \varepsilon^{\lambda}\eta^{\frac{\lambda}{2}}\mathbb{E}\|B(1)\|^{\lambda}. \quad \text{(S21)}$$

Consider the first term on the right side:

$$\|\hat{W}(j\eta) - \eta\nabla f(\hat{W}(j\eta))\|^{\lambda} = \|\hat{W}(j\eta) - \eta\nabla f(\hat{W}(j\eta))\|^{2\times\frac{\lambda}{2}}$$

$$= \Big(\|\hat{W}(j\eta)\|^{2} - 2\eta\langle\hat{W}(j\eta), \nabla f(\hat{W}(j\eta))\rangle + \eta^{2}\|\nabla f(\hat{W}(j\eta)\|^{2}\Big)^{\frac{\lambda}{2}}$$

$$\leq \Big(\|\hat{W}(j\eta)\|^{2} - 2\eta(m\|\hat{W}(j\eta)\|^{1+\gamma} - b) + \eta^{2}(2M^{2}\|\hat{W}(j\eta)\|^{2\gamma} + 2B^{2})\Big)^{\frac{\lambda}{2}}, \quad \text{(S22)}$$

where we have used assumption **A5** and Lemma S5. For $0 < \eta \leq \frac{m}{M^2}$,

$$2\eta m(\|\hat{W}(j\eta)\|^{1+\gamma} + 1) \geq 2\eta^{2}M^{2}\|\hat{W}(j\eta)\|^{2\gamma}. \qquad \text{(since } 1+\gamma > 2\gamma \text{ and } \eta m > \eta^{2}M^{2}\text{)}$$

Using this inequality we have

$$\|\hat{W}(j\eta) - \eta\nabla f(\hat{W}(j\eta))\|^{\lambda} \leq \Big(\|\hat{W}(j\eta)\|^{2} + 2\eta(b+m) + 2\eta^{2}B^{2}\Big)^{\frac{\lambda}{2}}$$

$$\leq \|\hat{W}(j\eta)\|^{\lambda} + (2\eta(b+m))^{\frac{\lambda}{2}} + 2^{\frac{\lambda}{2}}(\eta B)^{\lambda}. \qquad \text{(S23)}$$

Consider the case where $\lambda > 1$. By (S20) and (S23),

$$\left(\mathbb{E}\|\hat{W}((j+1)\eta)\|^{\lambda}\right)^{\frac{1}{\lambda}} \leq \left(\mathbb{E}\|\hat{W}(j\eta)\|^{\lambda} + (2\eta(b+m))^{\frac{\lambda}{2}} + 2^{\frac{\lambda}{2}}(\eta B)^{\lambda}\right)^{\frac{1}{\lambda}} + \varepsilon\eta^{\frac{1}{\alpha}}\left(\mathbb{E}\|L^{\alpha}(1)\|^{\lambda}\right)^{\frac{1}{\lambda}} + \varepsilon\eta^{\frac{1}{2}}\left(\mathbb{E}\|B(1)\|^{\lambda}\right)^{\frac{1}{\lambda}}$$

$$\leq \left(\mathbb{E}\|\hat{W}(j\eta)\|^{\lambda}\right)^{\frac{1}{\lambda}} + (2\eta(b+m))^{\frac{1}{2}} + 2^{\frac{1}{2}}\eta B + \varepsilon\eta^{\frac{1}{\alpha}}l_{\alpha,\lambda,d}^{\frac{1}{\lambda}} + \varepsilon\eta^{\frac{1}{2}}b_{\lambda,d}^{\frac{1}{\lambda}}$$

$$\leq \left(\mathbb{E}\|\hat{W}(0)\|^{\lambda}\right)^{\frac{1}{\lambda}} + (j+1)\left((2\eta(b+m))^{\frac{1}{2}} + 2^{\frac{1}{2}}\eta B + \varepsilon\eta^{\frac{1}{\alpha}}l_{\alpha,\lambda,d}^{\frac{1}{\lambda}} + \varepsilon\eta^{\frac{1}{2}}b_{\lambda,d}^{\frac{1}{\lambda}}\right).$$

For the case where $0 \leq \lambda \leq 1$, by (S21) and (S23),

$$\mathbb{E}\|\hat{W}((j+1)\eta)\|^{\lambda} \leq \mathbb{E}\|\hat{W}(j\eta)\|^{\lambda} + (2\eta(b+m))^{\frac{\lambda}{2}} + 2^{\frac{\lambda}{2}}(\eta B)^{\lambda} + \varepsilon^{\lambda}\eta^{\frac{\lambda}{\alpha}}l_{\alpha,\lambda,d} + \varepsilon^{\lambda}\eta^{\frac{\lambda}{2}}b_{\lambda,d}$$

$$\leq \mathbb{E}\|\hat{W}(0)\|^{\lambda} + (j+1)\left((2\eta(b+m))^{\frac{\lambda}{2}} + 2^{\frac{\lambda}{2}}(\eta B)^{\lambda} + \varepsilon^{\lambda}\eta^{\frac{\lambda}{\alpha}}l_{\alpha,\lambda,d} + \varepsilon^{\lambda}\eta^{\frac{\lambda}{2}}b_{\lambda,d}\right).$$

By using Lemma S6 and Corollary S1, we have the desired results.

$\square$

## S5 Details of the Simulations

We run the experiments for different values of the other parameters of the problem. The detailed settings of the parameters are as follows.

Figure 2(a) $d = 10$, $\alpha \in \{1.2, 1.4, 1.6, 1.8\}$, $\varepsilon = 0.1$, $\sigma = 1$, $a = 4 \times 10^{-4}$.

Figure 2(b) $d = 10$, $\alpha \in \{1.2, 1.4, 1.6, 1.8\}$, $\varepsilon \in \{10^{-3}, 10^{-2}, 10^{-1}, 10\}$, $\sigma = 1$, $a = 4 \times 10^{-6}$.

Figure 2(c) $d = 10$, $\alpha \in \{1.2, 1.4, 1.6, 1.8\}$, $\varepsilon = 0.1$, $\sigma \in \{10^{-2}, 10^{-1}, 1, 10\}$, $a = 4 \times 10^{-5}$.

Figure 2(d) $d \in \{10, 40, 70, 100\}$, $\alpha \in \{1.2, 1.4, 1.6, 1.8\}$, $\varepsilon = 0.1$, $\sigma = 1$, $a = 4 \times 10^{-4}$.

## S6 First Exit Times of Non-linear Dynamical Systems in $\mathbb{R}^d$ Perturbed by Multifractional Lévy Noise [8]

In this paper, the authors study a dynamical system which is perturbed by a $d$-dimensional Lévy process with $\alpha_i$-stable components. The authors investigate the exit behavior of the system from a domain $\mathcal{G}$ in the small noise limit and they prove that the system exits from the domain in the direction of the process with smallest $\alpha_i$. The main results of the paper are presented in Theorem 1, Proposition 1, Proposition 2 of the paper.