[Reviews · NeurIPS 2019]

Reviewer 1



Clarity: the paper is general clear, and concise regardless of the substantial technicalities (but see below). Paper organization is sensible in that too technical proofs are left to the appendix and the overarching proof strategy is put as a separate section. Originality: this line of research seems very recent, and this paper appears as a natural and timely continuation of [1], which empirically supports that gradient noise in SGD is heavy tailed. Quality and significance: I believe theoretical results by themselves are deep enough to judge this work as having high quality. This adds to the empirical validation, which are also insightful. I am only a bit concerned this work might be taken as an excuse for making theory, but losing connection with the broad machine learning community. [1] https://arxiv.org/abs/1901.06053

Reviewer 2



********* edit of review report after discussion ************ I have gone through all the other review reports, the authors' feedback, and the meta-reviewer's comments, as well as the discussion up to now. For Reviewer #1's concern about making theory, I tend to be open-minded since I can not find solid evidence that the paper is making theory only. For Reviewer #4's comment about the over-claim of the result the paper proved, my take is follows. First, for many problems, the true local minima enjoys the flat basin. A famous example I have is the following paper: McGoff, Kevin A., et al. "The Local Edge Machine: inference of dynamic models of gene regulation." Genome biology 17.1 (2016): 214. where the fitting to the gene expression level dynamics is expected to be flat near the true value, because the natural evolution tends to select the dynamics robust to the changes of the environment or external stimulations. Second, the authors have explained the motivation of using the Levy process to model the noise. As long as the motivation is valid, it is hard to say that the jumps are unusual. Based on the above reflection, I tend to maintain my previous rating of this paper, and am happy to vote for accepting the paper for publication. *************************************************************** Originality: The task is new and the methods, I believe, should be standard in the analysis of stochastic differential equations. This work is a novel combination of the techniques in stochastic differential equations, and the deep learning methodologies. The difference between this work and the previous works in literature is clear: the introduction of the S-alpha-S noise. To the best of our knowledge, the related works are sufficiently cited. Quality: Due to time restriction we have not got enough time to go through the proof. However, we are sure that all the claims are well supported by the theoretical analysis if the analysis is correct. The work presented in this paper is complete. The authors are careful and honest about evaluating both the strengths and weakness of their work. Clarity: The submission is well organized and clearly written. Since this is a theoretical paper, the proof provided in the supplementary materials form sufficient resource for readers to "reproduce" the results. Significance: The results are important. Since the analysis covers an important case of error distribution of which the existence is verified by experiments, the work presented in this paper has a big potential to be influential. I believe that other researchers and practitioners are likely to use the results of this paper. This submission addresses a difficult task in a better way than previous work, and advances the state-of-the-art results in a demonstratable way. The work provides unique theoretical analysis.

Reviewer 3



***************After Author response************ (1)-(2)-(4)-(5)-(6) were only recommendations and will of precisions. (3) was a mistake I made inferring from Theorem 1 that all results were only 1-dimensional. But the authors pointed out my mistake. Sorry for this lack of attention. The main concern I have is about the conclusions drawn from the result they prove. They deduce flatness near local minima from the fact that the basin of attraction of the latter is wide (from Theorem 2). I may be wrong but for me there is no reason that this happen even with the Holder-Gradient condition (even in R as they tried to prove in the rebuttal). Moreover, the big novelty with this model is the fact that the dynamics is not continuous anymore and seem to jump without taking care of the heights of the barriers. Do we observe these unusual jumps in practice ? I am sorry to show my skepticism because the paper is well-referenced and nice in general, I have only doubts on the fact that it model well Neural Networks. For all these reasons, I decide to keep my 5, as it is said in the description : Marginally below the acceptance threshold. I tend to vote for rejecting this submission, but accepting it would not be that bad. And if other reviewers and yourself are convinced by the model, I would be ok with it. ******************************************************* Originality: The paper is quite original and the new assumption on the noise model of SGD leads to a new continuous time limit SDE for the dynamic of SGD under which the exit time properties are fairly different (polynomial time versus usual exponential time). I appreciate the will of finding new good model for SGD that explains the practice. Quality-Clarity: The paper reads very clearly and is, despite its novel ideas and technical background for the ML community, fairly understandable. Moreover the continuous-time limit is explained and the result for the exit time of the discretized counterpart is related to the continuous one -which is not always the case in this literature. However, I am not totally convinced by the conclusions of the paper when affirming that as the time to exit from an interval is polynomial with respect to its width and does not depend on the height of barriers, SGD has a tendency to stay in wide minima bassins. To explain this phenomenon I would have expected a result depending on the flatness around local minima whereas the result is about every interval of width a (whether this is a neighbourhood of a local minimum or not). I would be glad to have more precisions about this in the rebuttal. Significance: see Contributions section.

[Author Response · NeurIPS 2019]

We thank all the reviewers for their time and effort to evaluate our paper. We appreciate that the reviewers find our
paper to be original, theoretically deep, well-organized, clearly written, complete, and our contributions to have a big
potential to be influential. We believe that we addressed all the raised issues. The detailed responses are given below.

**R1.** We thank the reviewer for the positive and insightful comments. As suggested by the reviewer, we will fix all the
minor issues: **(1)** We will replace $\bar{\varepsilon}$ with $\xi$ to prevent confusion. **(2)** We will define the concept of metastability in more
detail and provide more references. **(3)** To increase clarity, we will make A6 more concise by representing the constants
with Big-O notation and providing their explicit definitions in the supplementary document. **(4)** We agree that our most
important contribution is the fact that Theorem 2 enables the use of the metastability results for Lévy-driven SDEs for
their discretized counterpart. We will highlight this fact more clearly in a remark. Thank you for this suggestion. **(5)** In
our paper, we mainly aim at providing more understanding on the connections between SGD and wide minima. In the
current literature, the implication of better generalization by wide minima is still in an hypothesis phase and is a very
active research field on its own. Nevertheless, several empirical results have conformed with this phenomenon. We will
add a new paragraph on the connection between our results and generalization by discussing these points.

**R3.** We are grateful to the reviewer for the positive and encouraging comments. We will fix the typo as requested.

**R4.** We thank the reviewer for the detailed comments. We suspect that a simple misunderstanding about the scope
and the contributions of our paper might have influenced the opinions of the reviewer. We will now clarify this
misunderstanding and we hope this would help the reviewer to reconsider their score.

As acknowledged by Reviewers 1 and 3, modeling SGD as an SDE driven by Lévy motion has been already proposed
in a recent paper, Simsekli et al., ICML 2019 [6]. In that paper, the $\alpha$-stable noise assumption was indeed validated
empirically in various deep learning settings. Therefore, the empirical validation of the assumption has been in a way
validated in the community.

A clear limitation of [6] (also mentioned in their paper) is that the authors did not develop new theory and used existing
metastability properties of the **continuous-time** Lévy-driven SDEs (which we summarized in Section 2) as a proxy
for the **discrete-time** dynamics of SGD. Approximating SGD as a continuous-time SDE has already raised several
theoretical questions (as we mentioned in the beginning of page 3), since the behaviors of these two systems might
be significantly different. In our paper, we provided explicit conditions (Theorem 2) such that *the discretized process*
*inherits the metastability properties of its continuous-time limit*. Hence, our main contribution is to establish this
technically challenging theoretical result, which required us to first build a more general result about Lévy-driven SDEs
(Theorem 3) and use this result in a non-trivial way to relate the exit-times of the two processes (Theorem 2).

**Quality-Clarity:** We thank the reviewer for this insightful question. First, we would like to clarify that *our results only*
*hold for connected neighborhoods of a local minimum*. This can be verified by checking the assumption in Theorem
1 which explicitly requires the interval to be a neighborhood around a local minimum. Since we are relating the exit
time of the discretized process to the conclusions of Theorem 1, we automatically inherit this condition (to see this
clearly, Eqs 8-10 are explicitly defined using a neighborhood of local minimum $\bar{w}$). For exit times in $d$-dimensions,
we still require the set of interest to be a neighborhood of a local minimum (see A1-5 in [18]). Now, for simplicity
assume that we are in $\mathbb{R}$, and consider two local minima and define two intervals such that these intervals contain
exactly the basins around the local minima (similar to Figure 1 right in the paper). Then, if the exit time from Basin 1
is longer than Basin 2, it immediately implies that Basin 1 has a larger diameter. If we combine this fact with A3-5
(or the assumption in Theorem 1), which make sure that the function behaves globally regularly (gradient Hölder and
dissipative), we can directly deduce that Basin 1 will be more flat. The same argumentation can be made for $\mathbb{R}^d$ with
the careful construction of [18]. We agree that the connection with this notion and the other notions of flatness (e.g.
spectrum of the Hessian) is not immediate, yet we believe that there is an explicit link and it definitely deserves further
investigation. We also agree that this is a subtle point and we will clarify it by stating it explicitly.

**Improvements:** **(1)** We will define metastability in more detail as suggested. **(2)** The Brownian systems need
exponential time in the height of the basin (line 143, see also [17] Sec 3.1). We will explicitly define the exit
times as suggested. **(3)** We underline that our result is already for $\mathbb{R}^d$ with an explicit dependency on $d$. The multi-
dimensional version of Theorem 1 is available in [18] and makes a non-trivial connection between exit-times and the
dimension. However, that theorem would require us to introduce several technical constructs, which we could not
simply accommodate in our paper due to space limitations. We will summarize [18] in the supp. doc. for completeness.
**(4-5)** That is correct, as in [16], $x$ is the initial point of the process, taken uniformly in the interval $[-a, a]$. In the
multi-dimensional setting, the initial point $W(0)$ is in the neighbourhood of radius $a$, centered at the local minimum $\bar{w}$.
We will clarify these notations, thank you for pointing out. **(6)** The composite noise exhibits the same metastability
behavior as pure $\alpha$-stable noise, is more general, and is mathematically more convenient for our analysis (see line 232).
**(7)** In general, the law of the processes (6) and (7) are not the same (for any $\eta > 0$). However, one can show that (7)
converges (in law) to (6) when $\eta$ goes to zero (see [39]).

[Meta-Review · NeurIPS 2019]

The reviewers liked the paper and appreciated the authors feedback. The authors should implement all the recommendations from the reviewers in the final version of the paper.